# Qualitative and Quantitative Micro-CT Analysis of Natal and Neonatal Teeth

**DOI:** 10.3390/children9040560

**Published:** 2022-04-14

**Authors:** Mohammed Nadeem Bijle, Sreekanth Kumar Mallineni, James Tsoi

**Affiliations:** 1Paediatric Dentistry, Department of Clinical Sciences, College of Dentistry, Ajman University, Ajman 346, United Arab Emirates; m.bijle@ajman.ac.ae or; 2Center of Medical and Bio-Allied Health Sciences Research, Ajman University, Ajman 346, United Arab Emirates; 3Paediatric Dentistry, Department of Preventive Dental Science, College of Dentistry, Majmaah University, Al-Majmaah 11952, Saudi Arabia; 4Center for Transdisciplinary Research (CFTR), Saveetha Institute of Medical and Technical Sciences, Saveetha Dental College, Saveetha University, Chennai 600077, Tamil Nadu, India; 5Dental Materials Science, Discipline of Applied Oral Sciences and Community Dental Care, The University of Hong Kong, Hong Kong SAR, China

**Keywords:** infants, micro-computed tomography, mineral density, natal teeth, neonatal teeth

## Abstract

The objective of this study was to characterize natal and neonatal teeth using micro-computed tomography. A total of 4 natal and 11 neonatal teeth were used for the analysis. The reconstructed scans were assessed for the maximum enamel/dentin thickness and mineral density (MD). The scanned specimens were 3D reconstructed to qualitatively determine the surface topography. The dentin thickness was two-fold greater than enamel thickness for both natal and neonatal teeth (*p* < 0.05). The cervical third enamel MD remained undetermined in natal and neonatal teeth. The dentin MD at the cervical third for neonatal teeth was significantly lower than the incisal and middle third dentin (*p* < 0.05). Similarly, the dentin MD at the cervical third of neonatal teeth was significantly lower than the cervical third dentin MD of natal teeth (*p* < 0.05). Our qualitative analysis suggests that the cervical thirds of both natal and neonatal teeth are peculiar of an anomalous structure, with neonatal teeth showing an irregular outline. Under the conditions of the present study, it can be concluded that the neonatal teeth studied exhibited a distinguishable aberrant structure compared to the natal teeth. Therefore, the natal teeth unfold as a more organized, three-dimensional structure compared to the neonatal teeth.

## 1. Introduction

Usually, tooth eruption occurs when two-thirds to three-quarters of the root is developed [1]. While relative theories of tooth eruption might play a significant role in the eruption process, certain conditions deviate from this usual phenomenon, and the eruption timings of teeth are altered [2]. Amongst the conditions that feature the early eruption of teeth, clinicians have determined the presence of natal, neonatal, and precocious teeth in infants [3,4,5,6,7]. Natal teeth are tooth structures that prematurely erupt at birth, whereas neonatal teeth erupt during the first month post-birth [3,4]. Precocious teeth are referred to as erupted teeth beyond the first to the fifth month of infancy [8]. Often, representing the complement to primary teeth, mandibular incisors are the teeth affected, with a 1/10 probability of being supernumerary in origin. Irrespective of the time of eruption of these teeth, the structures of pre-deciduous teeth have been referred to as anomalous, sometimes presenting as shell-shaped teeth with no root structure [9]. However, very little is known in the literature about the differences between these pre-deciduous teeth, especially natal and neonatal teeth, which are commonly encountered in infants at or after birth.

The incidence of prematurely erupted natal or neonatal teeth in children is 1 in 2000 to 3500 live births, with a higher predilection for females [3]. While etiologies have been proposed, the most common anticipated reason for the premature eruption of teeth in infants is local disturbances in growth and development, including the superficial position of tooth germs [3,4]. Several reasons proposed for the premature eruption of natal/neonatal teeth include genetic factors, endocrinal disturbances, infection, and nutritional deficiencies [10]. Furthermore, natal or neonatal teeth could be syndromic and frequently associated with chondroectodermal dysplasia, pachyonychia congenita, Hallermann–Streiff syndrome, and Riga–Fede disease [11]. Whether an anomalous structure could potentiate eruptive changes in the local growth of natal/neonatal teeth remains unexplored.

With varying degrees of mineralization [9] and reported enamel hypoplasia/aplasia with natal/neonatal teeth, an alteration in the amelogenesis of the pre-deciduous teeth is suspected. Histological analysis of natal and neonatal teeth has suggested that the tooth structure is similar to an immature tooth [12,13]. Similarly, disturbances in tooth size, shape, color, and attachment to underlying soft tissues and alveolus have been reported [14]. Additionally, soft tissue pathologies, such as fibrous hyperplasia, granuloma, and cysts, are associated with the presence of natal/neonatal teeth [15,16,17,18]. This wide range of associated conditions might be peculiar to all or specific types of pre-deciduous teeth; this is unknown in the literature, as most such conditions are only reported for natal teeth and address inherent structural defects. Furthermore, it would be pertinent to investigate if any specific structural entities are distinctive to natal/neonatal teeth.

After a thorough literature review, we identified a paper that compared two extracted natal teeth with a sound exfoliated mandibular primary central incisor using micro-computed tomography (micro-CT) to assess tooth mineral density (MD), including enamel and dentin [9]. This study identified that the tooth MD for natal teeth was lower than the sound primary teeth. However, the sample size was too small to identify differences (if any) between the natal teeth under examination. Furthermore, none of the reported studies delineated differences between natal or neonatal teeth, which might interest clinicians. Since there is sparse information on natal and neonatal teeth in the dental literature, we planned to further characterize these teeth to uncover more information on the subject. Thus, this study’s objective was to characterize extracted natal and neonatal teeth using micro-CT scanning, thereby examining enamel/dentin mineral density, maximum thickness, and 3D reconstructed scans. Utilizing the variables of concern in the study objective, the tested hypothesis was that there is no difference in the physical and structural properties of natal and neonatal teeth estimated with micro-CT scanning.

## 2. Materials and Methods

### 2.1. Natal and Neonatal Teeth

A total of 4 natal and 11 neonatal teeth were included in the present study. The specimens were obtained following extraction in infants and stored in normal saline at 4 °C. The solution was renewed every week to maintain isotonicity. All teeth were stored and labeled with the recorded demographics for further analysis.

Following the procurement of the specimens and storage, the natal/neonatal teeth were prepared for scanning. The micro-CT scanning conditions are explained further. The tooth type, i.e., natal/neonatal, was kept anonymous to the primary operative investigator at all stages.

### 2.2. Micro-Computed Tomography Specimen Scanning

Specimens were scanned with reference hydroxyapatite phantoms of 0.25 and 0.75 g/cm^3^. The phantoms and the specimens were held stable on a customized thermocol enclosure that could be further inserted into the cylindrical receiver for scanning on the micro-CT scan (Skyscan™ 1172 X-ray Microtomograph, Bruker, Belgium) computer-controlled turntable. The scanning parameters were in line with previously published literature on micro-CT scans of enamel specimens [19,20].

Prior to each series of scans, flat field correction at 13 µm resolution was confirmed. Two specimens, irrespective of being natal/neonatal (following the allotted sequence), were assembled in the enclosure for respective scanning. The specimens were orientated in the cylindrical receiver so that the X-ray beam was projected perpendicular to the long axis of the teeth. The standardized scanning parameters for all specimens were source current of 100 µA, source voltage of 80 kV for Al filter of 0.5 mm, scanning at a rotation step of 1° for 360°, with an average frame of 2, and 10 arbitrary random movements. The specimens were exposed at 4840 msec, and all scanned files were received in TIFF format. After the scan, the specimens were stored in the designated media and conditions.

The scanned images were then reconstructed using NRecon v. 1.7.0.4 (SkyScan™, Bruker, Belgium) with the following variables: smoothing—1, ring artifact correction—10, and beam hardening correction—30. Once the images were reconstructed in 3D, each specimen was subjected to isolation using DataViewer v. 1.4.4.0 (SkyScan™, Bruker, Belgium). The specimens were first oriented per the long axis of the teeth and then sectioned in the frame comprising all content, including the adjacent reference phantom, for further assessment.

### 2.3. Maximum Enamel and Dentin Thickness

Several areas were identified and measured using the scale function in CTAn v. 1.16.1.0+ (SkyScan™, Bruker, Belgium). All measurements were made in mm format for standardization. The borders of enamel and dentin were identified using the observation and binary function of CTAn v. 1.16.1.0+ (SkyScan™, Belgium) to define appropriate thresholds.

Once the maximum thickness of the tissues was identified, the three most representative regions of enamel and dentin were marked and recorded for further analysis. A mean enamel/dentin thickness was derived, and further computed as a segregated enamel/dentin thickness for natal and neonatal teeth.

### 2.4. Enamel and Dentin MD Assessment

The quantitative enamel and dentin MD assessment was performed as per previous papers [19,20], referring to the hydroxyapatite phantoms. Calibration was performed for each specimen prior to determining the MD for the tissues. After importing the 3D reconstructed specimens into CTAn v. 1.16.1.0+ (SkyScan™, Belgium), the evident tissue structure was uniformly divided into three sections representing the incisal/occlusal, middle, and cervical third regions of the natal/neonatal teeth. For each tooth region (e.g., enamel/dentin), a stack of five 13 µm sections was designated as an area of interest for analysis. A uniform circular 100 µm (in diameter) region of interest was regulated with every stack to assess MD, representative of the area of interest. Based on five stacks, five such estimations of MD were made to identify a mean MD for the tooth region. In the end, mean total MD for each tooth tissue was determined for further statistical analysis.

### 2.5. Three-Dimensional Reconstruction of Micro-CT Scanned Specimens

The 3D reconstructed, micro-CT-scanned specimens were then imported at CT vol. v. 2.3.2.0 (Skyscan™, Bruker, Belgium). The specimens were oriented at the buccal, lingual, and proximal regions to define the structural outline of the natal/neonatal teeth qualitatively. For all teeth included in the study, high-resolution images in TIFF were made for further comparison. The most representative image for the natal and neonatal teeth was identified. A collage of images was prepared to be presented along with the quantitative data in the present paper.

### 2.6. Statistical Analysis

All the data obtained were entered into MS Office Excel (Microsoft 2019, Redmond, WA, USA) for further statistical analysis with SPSS v. 27 (IBM Statistics, Chicago, IL, USA).

The data on maximum enamel and dentin thickness with natal/neonatal teeth were analyzed with independent t-tests at two levels (e.g., within enamel/dentin data on natal/neonatal teeth and between enamel/dentin data in natal/neonatal teeth).

The enamel/dentin MD assessment data were analyzed using 2-way ANOVA with Bonferroni’s post hoc test (at all levels) that included two factors. Factor 1 was teeth type—natal/neonatal teeth; factor 2 was tooth region—incisal, middle, and cervical third.

The significance for all the data analyzed was set at *p* < 0.05.

## 3. Results

### 3.1. Maximum Enamel and Dentin Thickness

The data on the enamel/dentin thickness (in mm) are shown in Table 1. For each tissue (e.g., enamel/dentin), no significant difference in the maximum enamel/dentin thickness was discerned between natal and neonatal teeth (enamel: *p* = 0.205; dentin: *p* = 0.207). Conversely, the maximum dentin thickness for both natal and neonatal teeth was significantly greater than the enamel thickness (natal teeth: *p* = 0.002; neonatal teeth: *p* < 0.001). The maximum dentin thickness for both natal/neonatal teeth was around twice that of the enamel thickness. The results of the maximum enamel and dentin thicknesses suggest that the dentin thickness was around two-fold greater than the enamel thickness for both natal and neonatal teeth.

### 3.2. Mineral Density of Enamel and Dentin

The enamel MD of natal/neonatal teeth estimated at the incisal, middle, and cervical third is presented in Table 2. The enamel MD at the cervical third remained undetermined for both natal and neonatal teeth. The effect of factor 1 (tooth type: natal/neonatal teeth; *p* = 0.190) and the factor interaction (*p* = 0.926) was not statistically significant (*p* > 0.05), whereas the effect of factor 2 (tooth region: incisal, middle, or cervical third) on the enamel MD was statistically significant (*p* < 0.001). No significant difference was discerned between the natal/neonatal teeth and incisal/middle third enamel MD interaction (*p* = 0.926), while the difference was statistically significant (“~” to higher per case, given that undetermined has no value) to the undetermined data for both the natal and neonatal teeth (*p* < 0.001).

The data for estimated dentin MD for natal/neonatal teeth at the incisal, middle, and cervical thirds of the teeth are shown in Table 3. Unlike enamel MD, the MD for dentin with natal/neonatal teeth could be estimated at the incisal, middle, and cervical third. The effect of factor 1 (*p* = 0.047) and factor 2 (*p* < 0.001) on the dentin MD was statistically significant (*p* < 0.05), with no significant difference noted for the factor interaction (*p* = 0.468). For natal teeth, no significant difference was observed between the incisal, middle, and cervical third dentin MD (*p* > 0.05). Conversely, for neonatal teeth, the dentin MD at the cervical third was significantly lower than the incisal/middle third (*p* < 0.05). Furthermore, the dentin MD at the cervical third for natal teeth was significantly higher than the neonatal teeth (*p* = 0.046). In contrast, no significant difference could be discerned between the dentin MD of natal and neonatal teeth at each incisal and middle third of the teeth.

### 3.3. Three-Dimensional Reconstruction of Natal/Neonatal Teeth

Figure 1 images represent the 3D reconstructed natal teeth, post-micro-CT scanning. The representative images for 3D reconstructed neonatal teeth are presented in Figure 2. Figure 1 and Figure 2 illustrate that the cervical third enamel structure of both natal and neonatal teeth appears atypical to the incisal and middle third regions. Lobe-like structures are evident with natal and neonatal teeth. The natal tooth (Figure 1) appears congruous in outline, while the neonatal tooth (Figure 2) has bouts of structure defacements, suggestive of a disorganized structural development.

The qualitative analysis of the 3D reconstructed, micro-CT-scanned natal/neonatal teeth suggest that the cervical third of natal/neonatal teeth is peculiar, with an anomalous structure, where neonatal teeth show an irregular outline.

Overall, the results of the present study demonstrate that natal teeth with higher dentin MD (specific to the cervical third dentin), similar enamel MD and enamel/dentin thickness, and a more congruent tooth structure outline than neonatal teeth, are suggestive of a systematically organized developmental tooth structure.

## 4. Discussion

The present study aimed to characterize extracted natal and neonatal teeth using micro-CT and to reconstruct the micro-CT-scanned images to estimate the enamel/dentin thickness, MD, and qualitatively analyze the structural topography. The results of the study suggest differences in the dentin MD and tooth structure between natal and neonatal teeth. Given the results of the present study, the null hypothesis for the study was rejected. It was discerned that there are evident differences between natal and neonatal teeth specific to the cervical third dentin and the tooth structure outline, as qualitatively analyzed using micro-CT-enabled, 3D reconstructed natal/neonatal teeth specimens.

Pre-deciduous (natal/neonatal) teeth are prematurely erupted teeth in infants that are affected by either chronological alteration or eruption timings, with the latter being dominant in occurrence. Concerns have been raised that such teeth need to be extracted early due to the possible aspiration risks in infants, which have been ruled out by previously published studies based on a lack of evidence [9,21,22]. Other relevant reasons for extracting natal/neonatal teeth are hypermobility, tongue ventral surface ulceration, root agenesis, and feeding difficulties [10,11]. In addition, some papers have advocated for the retention of natal teeth with thorough, extensive preventive regimes for long-term sustenance [14]. Although previous studies have outlined that natal/neonatal teeth appear anomalous and immature, their retention in the oral cavity is not justified well by the tooth structure [12,13,14]. From the present study results, it can be inferred that the structures of the natal teeth were well-organized when compared to the neonatal teeth, even though a previous study outlined that the natal tooth has a lesser MD than the usually erupted primary teeth [9]. Therefore, the concern and focus should be shifted to neonatal teeth, which project structural deficiencies and long-term retention problems. However, the focus shift should be dealt with cautiously, as the size and lesser MD with natal teeth, compared to the usual primary teeth in the oral cavity, should still be considered.

The enamel and dentin thicknesses determined in natal/neonatal teeth in the present study reveals that the dentin thickness was two-fold greater than the enamel in both natal and neonatal teeth. A previous study that estimated enamel and dentin thicknesses with micro-CT also outlined similar findings, whereby the dentin thickness was around two-fold greater than the enamel thickness for soundly extracted premolar teeth [23]. Therefore, it can be inferred that natal/neonatal teeth maintain a similar enamel and dentin ratio as the sound teeth that erupt in the oral cavity. The rationale for such an observation could be attributed to the origin of the teeth most commonly belonging to the usual complement of the primary teeth (around 90%), as opposed to supernumerary teeth [9]. Therefore, in the present study, the teeth under investigation had origins in the usual complement of primary teeth instead of supernumerary teeth, which was not ruled out otherwise by a loss to follow-up.

The enamel MD between the natal and neonatal teeth (at incisal and middle thirds) did not differ. In contrast, the dentin MD specific to the cervical dentin with natal teeth had higher dentin MD than the neonatal teeth. It is significant to note that the cervical third enamel MD was undetermined, as not all specimens had identifiable enamel in the region. Moreover, surface topographical analysis of the natal/neonatal teeth specimens revealed that the natal teeth had a more organized structure than the neonatal teeth, with evident enamel hypoplasia/aplasia at the cervical third of natal and neonatal teeth. Areas affected by enamel hypoplasia other than cervical third regions were observed with neonatal teeth during the qualitative structural analysis of 3D reconstructed specimens. The observations made in this study advocate that neonatal teeth highlight more precisely anomalous areas than natal teeth. It would be interesting to further investigate if the delay in eruption with neonatal teeth compared to natal teeth is related to the anomalous structure, which fails to potentiate early eruptive changes even if a favorable tooth germ position is present. However, studies to identify this theorized pattern with a larger sample size are needed to confirm the phenomenon.

The dentin MD at the cervical third was lower than the incisal/middle third for neonatal teeth. Upon close observation of the data, it was estimated that more than 50% of neonatal teeth had absolutely no identifiable enamel in the cervical third region compared to merely 25% (i.e., one specimen) of natal teeth. Prior research explained that amelogenesis is secondary to dentinogenesis, thus signaling defects during dentin formation led to the changes in enamel hypoplasia/aplasia in the cervical third of these teeth [24]. However, the defects might have been more pronounced in the neonatal teeth compared to natal teeth. One study explained that natal and neonatal teeth had irregular dentin deposition at the cervical thirds of the teeth, which conforms to the results of the neonatal teeth in the present study [13]. Further, it has been explained that dentin defects in the cervical region stimulate the degeneration of Hertwig’s root sheath, thereby affecting root development. Despite the retention of natal teeth for around a year, root development was not evident in a reported case [14]. The previous explanations suggest that, irrespective of natal or neonatal teeth, the cervical third regions of such teeth are growth-affected and thus appear anomalous. The present study results further add that dentin in the cervical third region might not be affected in natal teeth compared to neonatal teeth.

The present study investigations were conducted on 4 natal and 11 neonatal teeth using micro-CT scanning and related reconstruction-assisted assessments, which were comprehensively performed. The study results add to the existing literature on the potential differences between natal and neonatal teeth. However, more conclusive remarks on the inferences of the present study could be made with a larger sample size of natal and neonatal teeth undergoing similar characterization and assessment as in the present study. Furthermore, chemical analysis of the teeth might provide more information on the chemical interplay in such teeth; however, limitations arise in identifying information in the individual regions. Additionally, future studies can determine if natal/neonatal teeth exhibit differences while being either supernumerary or from the usual complement of primary dentition in origin. Thus, the present study establishes a platform for future studies on natal and neonatal teeth with a special mention of the possible evolutionary changes that might have occurred with their anomalous appearance in the current era.

## 5. Conclusions

Under the conditions of the present study, it can be concluded that neonatal teeth exhibit a distinguishable, aberrant structure compared to natal teeth. Therefore, natal teeth unfold as a more organized, three-dimensional structure compared to neonatal teeth. With further studies, long-term retention of natal teeth, as opposed to neonatal teeth, can be revisited.

## Figures and Tables

**Figure 1 children-09-00560-f001:**
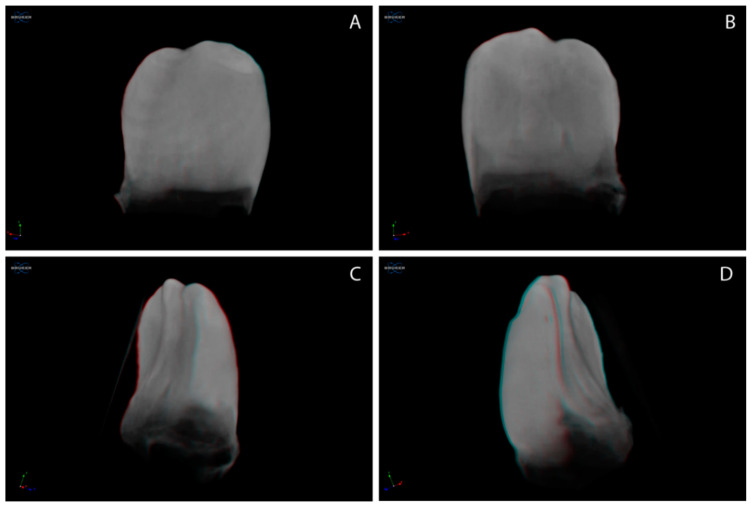
Representative 3D reconstructed natal tooth. (**A**) buccal view; (**B**) palatal view; (**C**,**D**) proximal views.

**Figure 2 children-09-00560-f002:**
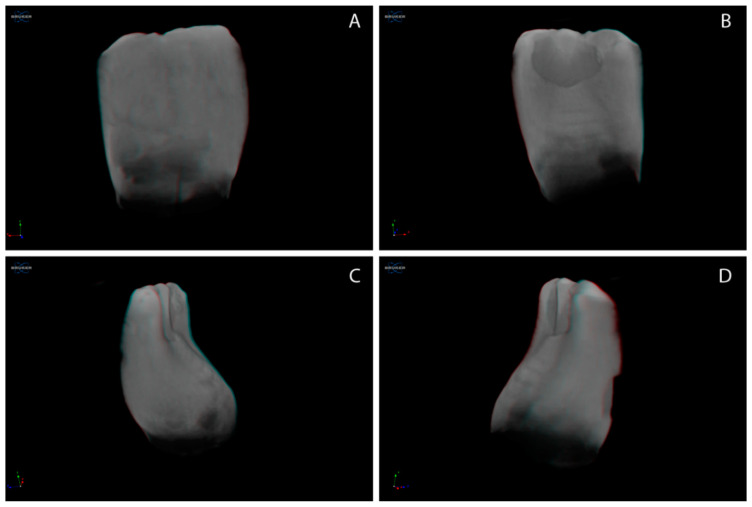
Representative 3D reconstructed neonatal tooth. (**A**) buccal view; (**B**) palatal view; (**C**,**D**) proximal views.

**Table 1 children-09-00560-t001:** Maximum enamel and dentin thicknesses (in mm) in natal/neonatal teeth.

Thickness (in mm)			
Teeth	Enamel	Dentin	*p*-Value
Natal (*n* = 4)	0.26 ± 0.07 ^a^	0.50 ± 0.06 ^b^	0.002
Neonatal (*n* = 11)	0.31 ± 0.04 ^A^	0.48 ± 0.08 ^B^	<0.001

No significant difference was observed between natal and neonatal teeth for maximum enamel/dentin thickness (*p* > 0.05). Uppercase and lowercase English letters represent significant differences between enamel and dentin maximum thicknesses for natal and neonatal teeth.

**Table 2 children-09-00560-t002:** Enamel mineral density (in g/cm^3^) at incisal, middle, and cervical thirds of natal/neonatal teeth.

Enamel Mineral Density (Mean ± SD) in g/cm^3^			
Teeth	Incisal 1/3rd	Middle 1/3rd	Cervical 1/3rd
Natal (*n* = 4)	1.79 ± 0.27 ^a,1^	1.71 ± 0.20 ^a,I^	Undetermined ^b,α^
Neonatal (*n* = 11)	1.71 ± 0.17 ^A,1^	1.53 ± 0.18 ^A,I^	Undetermined ^B,α^

Two-way ANOVA with Bonferroni’s post hoc test at all levels was applied. Factor 1—natal/neonatal teeth (*p* = 0.190); Factor 2—incisal/middle/cervical third of the tooth (*p* < 0.001); Factor interaction (*p* = 0.926). Lower (a,b)/uppercase (A,B) letters indicate differences within each row. (1), (I), and (α) show differences within each column.

**Table 3 children-09-00560-t003:** Dentin mineral density (in g/cm^3^) at incisal, middle, and cervical thirds of natal/neonatal teeth.

Dentin Mineral Density (Mean ± SD) in g/cm^3^			
Tooth Type	Incisal 1/3rd	Middle 1/3rd	Cervical 1/3rd
Natal (*n* = 4)	1.25 ± 0.14 ^a,1^	1.21 ± 0.12 ^a,I^	1.09 ± 0.13 ^a,α^
Neonatal (*n* = 11)	1.23 ± 0.12 ^A,1^	1.12 ± 0.11 ^A,I^	0.95 ± 0.12 ^B,β^

Two-way ANOVA with Bonferroni’s post hoc test at all levels was applied. Factor 1—natal/neonatal teeth (*p* = 0.047; natal > neonatal); Factor 2—incisal/middle/cervical third of the tooth (*p* < 0.001); Factor interaction (*p* = 0.468). Lower(a)/uppercase (A,B) letters demonstrate differences within each row. (1), (I), and (α,β) show differences within each column.

## Data Availability

The data presented in this study are available on request from the corresponding author. The data are not publicly available due to ethical concerns.

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
