# Peer review of "Qualitative and Quantitative Micro-CT Analysis of Natal and Neonatal Teeth"

_children, 2022, doi:10.3390/children9040560_

Round 1
Reviewer 1 Report
Dear Sirs, thank you for the opportunity to review the paper. It is a novel paper, although it has some flaws that should be corrected.
1. In order to keep the adequate proportion of self-cites to cites in this article, please consider widening the literature, eg.
-
- Vahabzadeh Z, Hashemi ZM, Nouri B, Zamani F, Shafiee F. Salivary enzymatic antioxidant activity and dental caries: A cross-sectional study. Dent Med Probl. 2020;57(4):385–391. doi:10.17219/dmp/126179 - when writing about the use of the tooth pastes and prevention of caries
- Paradowska-Stolarz A. MSX1 gene in the etiology orofacial deformities. Postepy Hig Med Dosw (Online). 2015 Dec 31;69:1499-504. - in relation to the genetics in presence of teeth anomalies
- E. Magitot, “Anomalies in the erupton of the teeth in man,” The British Journal of Dental Science, vol. 26, pp. 640–641, 1883. - as a possibly firstly described problem of natal and neonatal teeth
- Nardi GM, Grassi R, Ndokaj A, et al. Maternal and Neonatal Oral Microbiome Developmental Patterns and Correlated Factors: A Systematic Review-Does the Apple Fall Close to the Tree?. Int J Environ Res Public Health. 2021;18(11):5569. - for discussion session
- Sarul M, Kawala B, Kozanecka A, Łyczek J, Antoszewska-Smith J. Objectively measured compliance during early orthodontic treatment: Do treatment needs have an impact? Adv Clin Exp Med. 2017 Jan-Feb;26(1):83-87
- Nahajowski M, Hnitecka S, Antoszewska-Smith J, Rumin K, Dubowik M, Sarul M. Factors influencing an eruption of teeth associated with a dentigerous cyst: a systematic review and meta-analysis. BMC Oral Health. 2021 Apr 7;21(1):180. doi: 10.1186/s12903-021-01542-y. - lines 63-64
- In general, the citations should be widened to at least 30-35 positions, if the Authors want to keep their autocitations
- Please, provide information from where did you know if teeth were natal or neonatal (interview, observations) and when (which day of life) were they extracted, if the informaiton is provided
- Skyskan is a reserved name - please, add this sign when the name is mentioned
- Did you get the information wheather the teeth were supernumerary? Please, discuss that condition with anomalies in the structure, especially with supernumerary teeth
- Please, add the limitations of the study (see the previous information).
- Please, add information if you have a bioethical permits to present this study (please, give reference number if yes)
After completion of the information, please resubmit the paper for further review.
Author Response
Reviewer 1 comments:
- Dear Sirs, thank you for the opportunity to review the paper. It is a novel paper, although it has some flaws that should be corrected.
Response 1
Many thanks for considering our paper novel. We appreciate your efforts in reviewing the paper and have worked through your comments, as and where applicable.
- In order to keep the adequate proportion of self-cites to cites in this article, please consider widening the literature, eg.
- Vahabzadeh Z, Hashemi ZM, Nouri B, Zamani F, Shafiee F. Salivary enzymatic antioxidant activity and dental caries: A cross-sectional study. Dent Med Probl. 2020;57(4):385–391. doi:10.17219/dmp/126179 - when writing about the use of the tooth pastes and prevention of caries
- Paradowska-Stolarz A. MSX1 gene in the etiology orofacial deformities. Postepy Hig Med Dosw (Online). 2015 Dec 31;69:1499-504. - in relation to the genetics in presence of teeth anomalies
- Magitot, “Anomalies in the erupton of the teeth in man,” The British Journal of Dental Science, vol. 26, pp. 640–641, 1883. - as a possibly firstly described problem of natal and neonatal teeth
- Nardi GM, Grassi R, Ndokaj A, et al. Maternal and Neonatal Oral Microbiome Developmental Patterns and Correlated Factors: A Systematic Review-Does the Apple Fall Close to the Tree?. Int J Environ Res Public Health. 2021;18(11):5569. - for discussion session
- Sarul M, Kawala B, Kozanecka A, Łyczek J, Antoszewska-Smith J. Objectively measured compliance during early orthodontic treatment: Do treatment needs have an impact? Adv Clin Exp Med. 2017 Jan-Feb;26(1):83-87
- Nahajowski M, Hnitecka S, Antoszewska-Smith J, Rumin K, Dubowik M, Sarul M. Factors influencing an eruption of teeth associated with a dentigerous cyst: a systematic review and meta-analysis. BMC Oral Health. 2021 Apr 7;21(1):180. doi: 10.1186/s12903-021-01542-y. - lines 63-64
- In general, the citations should be widened to at least 30-35 positions, if the Authors want to keep their autocitations
Response 2
Many thanks for your suggestion. We have included references based on your suggestion. The following published articles have been cited further in the list which appear relevant to the contributors.
References:
American Academy of Paediatric Dentistry. Perinatal and infant oral health care. Ref. Man. Pediatr. Dent. Chicago, III. Am. Acad. Pediatr. Dent. 2021, 262–6.
American Academy of Paediatric Dentistry. Periodicity of examination, preventive dental services, anticipatory guidance/counseling, and oral treatment for infants children and adolescents. Ref. Man. Pediatr. Dent. Chicago, III. Am. Acad. Pediatr. Dent. 2021, 241–51.
Magitot, E. Anomalies in the erupton of the teeth in man. Br. J. Dent. Sci. 1883, 26, 640–641.
Nahajowski, M.; Hnitecka, S.; Antoszewska-Smith, J.; Rumin, K.; Dubowik, M.; Sarul, M. Factors influencing an eruption of teeth associated with a dentigerous cyst: a systematic review and meta-analysis. BMC Oral Health 2021, 21, 1–11, doi:10.1186/s12903-021-01542-y.
Kindly note that the count is yet to 24 papers cited in the manuscript which are relevant to the topic in discussion. In addition, we have removed two self-cites on the micro-CT scan and analysis methodology while retaining two recent papers. The recent papers cited by the authors on [19,20] are addressing the characterization technique which is an expertise of the research team and whereby the methodology is well described.
- Please, provide information from where did you know if teeth were natal or neonatal (interview, observations) and when (which day of life) were they extracted, if the informaiton is provided
Response 3
The teeth were classified as natal or neonatal based on clinical observations as per the description mentioned on Lines 36-38. The information on extraction of the natal/neonatal teeth was not recorded as the data would be different than the study objectives. Furthermore, as mentioned on Lines 90-92, the information on classified natal/neonatal teeth was kept anonymous to the primary investigator until the data analysis whereby the focus was mainly on the objectives of the study.
- Skyskan is a reserved name - please, add this sign when the name is mentioned
Response 4
Thank you for your comment. We have now included the sign – ‘™’ where Skyscan is mentioned.
- Did you get the information wheather the teeth were supernumerary? Please, discuss that condition with anomalies in the structure, especially with supernumerary teeth
Response 5
We did not get the information if the teeth were supernumerary or from the usual complement of the primary dentition. This has been addressed in Lines 248-253. Moreover, any additional information on how different would the natal/neonatal teeth with supernumerary or primary dentition appear is a matter of further investigation which is yet to be undertaken and definitely a hypothesis to be tested. Many thanks for your suggestion for future research.
- Please, add the limitations of the study (see the previous information).
Response 6
The study limitations are well addressed in the last paragraph of the discussion specifically at Lines 288-293.
- Please, add information if you have a bioethical permits to present this study (please, give reference number if yes)
Response 7
Yes, certainly we have a bioethics permission and the details of which are now added to the manuscript. Please see Lines 308-310.
- After completion of the information, please resubmit the paper for further review.
Response 8
The comments from the Reviewer 1 are addressed in this rebuttal and the manuscript is revised as applicable. We hope that the manuscript will be acceptable to the reviewers in the current form.

Reviewer 2 Report
Dear Authors,
Thank you for submitting your valuable work to the journal. The topic of the paper is intersting, but there are some comments I would make in order to
improve its scientific accuracy.
- Please rephrase and reorganize English language style: it is often informal and lacks scientific rigour
- Please add Inclusion/Exclusion criteria for teeth selection
- Please state whether a power statistical test was performed, and its result if performed
- Please add Clinical consequences and perspectives in the Discussion Section
- Please rephrase Conclusion section for improved comprehension to a single idea/phrase
- Please check English grammar and vocabulary
We look forwards to receiving the revised version!
Thank you!
Author Response
Reviewer 2 comments:
- Dear Authors,
Thank you for submitting your valuable work to the journal. The topic of the paper is intersting, but there are some comments I would make in order to improve its scientific accuracy.
Response 1
Many thanks for your valuable time to review the paper. We appreciate your efforts.
- Please rephrase and reorganize English language style: it is often informal and lacks scientific rigour
Response 2
We have gone through the paper content once again to address scientific rigour, English syntax and grammer, and organisation of contents. Minor changes are made to the content as agreed by all the contributors of the manuscript.
- Please add Inclusion/Exclusion criteria for teeth selection
Response 3
The teeth were selected based on the convenience sampling technique as the condition prevalence is 1 in 2000-3500 live births (Please see Lines 46-47). As addressed to the Reviewer 1 Comment 3, Response 3; the teeth were classified based on the criteria mentioned for natal/neonatal teeth which are well known in the literature.
- Please state whether a power statistical test was performed, and its result if performed
Response 4
The power statistical test was not performed for the present study as it is purely observational in nature with no interventional component. Even post-hoc power analysis might not relate well to the study characteristics. Hence, no power test was performed either a priori or post-hoc.
- Please add Clinical consequences and perspectives in the Discussion Section
Response 5
A definite clinical consequence and perspective on the subject is yet to be discerned. However, details have been provided in paragraph 1, 3, & 4 of the discussion section which explains the clinical consequences of the neonatal teeth exhibiting structural deficiencies that might lead to long-term retention concerns, delay in eruption of the teeth (as neonatal teeth erupt later than natal teeth is scheduled for eruption), and tooth formation signaling defects which has a molecular perspective to be identified.
- Please rephrase Conclusion section for improved comprehension to a single idea/phrase
Response 6
The conclusion is now re-phrased for improved comprehension based on the structural properties identified with natal and neonatal teeth. The revised conclusion is as follows:
Lines 299-301: Under the conditions of the present study, it can be concluded that the neonatal teeth exhibited a distinguishable aberrant structure than the natal teeth. Therefore, the natal teeth unfold as an organized three-dimensional structure than the neonatal teeth.
- Please check English grammar and vocabulary
Response 7
We have addressed this comment at Reviewer 2 Comment 2, Response 2. We hope it is acceptable.
- We look forwards to receiving the revised version!
Thank you!
Response 8
The revised version of the manuscript is now ready for your kind perusal. We hope that the revised paper is now acceptable for publication in Children – Special Issue of Paediatric Oral Health.

Round 2
Reviewer 1 Report
Dear Sirs,
Thank you for answers to all of my suggestions. Here are several ones more:
- Please add to the discussion, that for the future studies / furhter investigations you should check "how different would the natal/neonatal teeth with supernumerary or primary dentition appear"
- Please, provide English checking by MDPI, because I still have some doubts.
Best regards!
Author Response
Reviewer 1 comments:
Dear Sirs,
Thank you for answers to all of my suggestions. Here are several ones more:
- Please add to the discussion, that for the future studies / furhter investigations you should check "how different would the natal/neonatal teeth with supernumerary or primary dentition appear"
Response 1
Thank you for your comment. We have included the suggested information in the revised manuscript on Lines 289-291.
- Please, provide English checking by MDPI, because I still have some doubts.
Response 2
We requested MDPI for their service for an English language check which is enclosed along. The editing by MDPI was verified to accept/reject changes and the revised manuscript is presented for your kind perusal.
Best regards!
Reviewer 2 Report
The authors have addressed all given suggestions accordingly. Thank you for your effort!
Author Response
Reviewer 2 comments:
- The authors have addressed all given suggestions accordingly. Thank you for your effort!
Response 1
Thank you for your efforts to re-review the manuscript. We appreciate your valuable feedback.